# Procedural Fairness Failures in RLHF from Preference Averaging

**M P V S Gopinadh**     **Karthik Kamuju**     **Kummari Avinash**     **Muppana John Joshua**
**Srinivasa Raju Rudraraju**
Vishnu Institute of Technology
`mpavangopinadh@gmail.com`

## Abstract

Reinforcement Learning from Human Feedback (RLHF) aggregates heterogeneous preferences into a single reward model, assuming preference homogeneity. When preferences are heterogeneous, this aggregation induces a procedural fairness failure where majority preference groups dominate reward learning while minority preferences are systematically under-represented. This work defines procedural fairness in alignment as preserving distinct preference signals during reward modeling and shows that standard RLHF violates this via preference averaging. Preference-Aware RLHF (PA-RLHF) is introduced, separating optimization across preference modes at the reward learning stage. In a controlled setting, PA-RLHF improves overall alignment accuracy from 46.9% to 67.9% and reduces the fairness gap between best and worst aligned groups from 15.9 to 9.6 percentage points. These results show that procedural fairness failures in alignment can arise from structural design choices in reward learning, even in controlled, noise-free settings, with direct implications for large language models and agentic systems, where biased reward models can compound inequities across sequential decisions.

## 1 Introduction

Large language models are increasingly deployed as general-purpose systems serving diverse users and use cases. As they interact with heterogeneous populations, expectations for their behavior vary substantially. Preferences differ across dimensions such as verbosity, reasoning depth, and style. These differences are often incompatible and not merely noisy. Contemporary alignment pipelines, most notably Reinforcement Learning from Human Feedback (RLHF), aggregate such heterogeneous feedback into a single reward model, implicitly assuming that user preferences admit a shared normative target Ziegler et al. (2019); Ouyang et al. (2022).

Standard RLHF optimizes a single reward model over aggregated preference data, causing preference influence to scale with dataset frequency. Alignment behavior thus reflects prevalence alongside content, enabling majority preferences to dominate while systematically under-optimizing minority groups. This constitutes a procedural fairness failure, as the alignment objective itself disadvantages minority preferences even when groups are internally consistent. In agentic systems, where reward models guide multi-step planning and action, such early aggregation can repeatedly bias subsequent decisions, leading minority preferences to be persistently ignored over time.

Prior work on preference-based alignment has primarily focused on personalization and controllability. Latent-variable preference models and preference-conditioned reward learning enable adaptation to individual users without explicit labels Poddar et al. (2024); Gong et al. (2025). Other approaches explore inference-time control over multiple preference dimensions through conditional optimization Guo et al. (2024). While effective at adapting models to users, these approaches do not address how the training objective itself prioritizes some preferences over others when preferences conflict Kirk et al. (2023).

Recent work has largely evaluated fairness at the level of model outputs, with limited attention to how alignment procedures distribute optimization pressure across preference groups. As a result, the role of preference aggregation in producing systematic group-level misalignment remains under examined Xie et al. (2025); Alabi & Wick (2024).

This work identifies preference averaging in RLHF as a structural source of procedural unfairness that leads to systematic group-level misalignment. To address this, we introduce Preference-Aware RLHF (PA-RLHF), which separates reward optimization across preference modes to preserve heterogeneous preference signals during training. In a controlled experimental setting, we show that this approach reduces fairness gaps and improves alignment accuracy across preference groups, demonstrating that procedural design choices in reward learning can directly shape group-level alignment outcomes. The analysis focuses on how aggregating heterogeneous preferences in RLHF shapes group-level alignment, and whether separating reward learning improves fairness.

## 2 Methodology

PA-RLHF intervenes at the reward learning stage of RLHF by preventing heterogeneous preferences from collapsing into a single optimization objective. The alignment process preserves distinct preference signals during reward learning.

### 2.1 Experimental Setting and Preference Data

This study evaluates PA-RLHF in a controlled setting isolating preference aggregation effects under heterogeneity. The dataset contains 971 pairwise comparisons from 60 simulated raters across 20 prompts. Raters are programmatically assigned to three groups of 20, each with a distinct preference profile: concise responses (20-35 words, 85% within-group consistency), detailed responses (60-90 words, 85% consistency), and technical/formal responses (40-60 words, 80% consistency). Each rater evaluated 15-18 randomly sampled prompts. This controlled simulation isolates preference aggregation effects from annotation noise and demographic confounds.

Repeated preference comparisons reveal three latent preference groups characterized by internally consistent yet mutually conflicting alignment criteria—for example, preferences favoring concise answers over detailed explanations, or technical rigor over stylistic flexibility. Each feedback instance consists of a prompt, two candidate responses, and a binary preference label. Data are split by raters using a 75/25 train-test partition. Ground-truth group assignments are used exclusively for evaluation. The base language model is held fixed throughout reward learning to ensure that observed effects arise from preference aggregation over representation learning.

### 2.2 Preference-Aware RLHF Pipeline

PA-RLHF modifies reward learning by explicitly separating optimization across inferred preference modes. Each preference comparison is embedded using fixed sentence-level representations over prompt–response pairs.

Representations are clustered into $k = 3$ preference modes, selected via silhouette score sweep over $k = 2$–$6$ on held-out data (silhouette $= 0.199$, ARI $= 0.443$). The low silhouette score reflects that preference groups differ stylistically, not semantically, producing soft geometric boundaries in embedding space. ARI $= 0.443$ confirms meaningful alignment with ground-truth group structure, supporting $k = 3$ as a principled choice. Fixing the number of modes isolates procedural effects from errors in preference discovery. Adaptive mode selection is deferred to future work. A separate reward model is trained for each mode using only the feedback assigned to that cluster, with a simple parametric model to isolate procedural effects and avoid confounding model capacity. Reward learning proceeds independently within each mode, ensuring that optimization updates for one preference group are not influenced by the relative frequency of other, potentially conflicting preferences. Clustering is used only to separate preference signals; the observed improvements arise from eliminating cross-group interference during reward learning, not from clustering itself.

Prompt-response pairs are embedded using all-MiniLM-L6-v2 (SBERT, 384-dim, frozen weights). User feature vectors are the mean of preferred-response embeddings concatenated with a response-length scalar (385-dim), normalized to zero mean and unit variance. Clustering uses K-Means with 20 restarts (seed 42). The reward model is Logistic Regression trained on the concatenation of the prompt embedding and the difference between response embeddings (768-dim total), approximating Bradley-Terry pairwise preference.

Alignment is evaluated by scoring candidate responses using the corresponding preference-specific reward model, without performing full policy optimization. This separation ensures that minority preference modes receive comparable optimization attention to majority modes during reward learning, preventing dominance through aggregation. By separating reward optimization across preference modes, PA-RLHF ensures that no group's influence is diminished solely due to lower prevalence, enforcing procedural fairness directly at alignment time.

## 3 RESULTS

Procedural fairness is assessed through group-level alignment accuracy disaggregated by preference group (Table 1). Alignment accuracy measures agreement with group-specific preference judgments. Standard RLHF exhibits substantial group-level imbalance despite moderate aggregate performance. While overall alignment accuracy reaches 46.9%, the most aligned preference group achieves 56.2% accuracy, whereas the least aligned groups reach only 40.3-41.2%, producing a fairness gap of 15.9 percentage points between best and worst-aligned preference groups.

PA-RLHF directly addresses this outcome by separating reward learning across preference modes, reducing the fairness gap to 9.6 percentage points (a 40% reduction). While PA-RLHF substantially reduces the fairness gap, the residual 9.6 pp difference suggests that clustering-based separation alone does not eliminate all group-level misalignment, motivating further procedural refinements. Minority preference groups experience large absolute gains in alignment accuracy (+27.6 and +32.8 percentage points), while the majority group improves only marginally (+7.3 percentage points). These results show that aggregation reallocates optimization pressure toward majority preferences as a consequence of objective construction, producing predictable minority under-alignment even in noise-free settings.

Table 1: Alignment accuracy by preference group under standard RLHF and PA-RLHF

| Preference Group | Baseline | PA-RLHF | Improvement |
|---|---|---|---|
| Overall Accuracy | 46.9% | 67.9% | +21.0 pp |
| Mode 0 (Minority) | 41.2% | 68.8% | +27.6 pp |
| Mode 1 (Majority) | 56.2% | 63.5% | +7.3 pp |
| Mode 2 (Minority) | 40.3% | 73.1% | +32.8 pp |
| Fairness Gap | 15.9 pp | 9.6 pp | -6.3 pp (40%) |

## 4 DISCUSSION

Procedural fairness in alignment requires that no preference group is systematically disadvantaged by the construction of the learning objective itself. Standard RLHF enforces a single normative objective over inherently pluralistic preferences, producing a procedural fairness failure that systematically under-aligns minority preference groups. Downstream interventions address symptoms after deployment, whereas the inequities observed here originate upstream, at design time, through the construction of the learning objective itself. Preserving preference heterogeneity during reward learning offers a principled alternative, suggesting that equitable alignment requires explicit structural accommodation of diversity rather than reliance on averaged objectives that obscure minority signals. The goal of this work is diagnostic and scope-limited: to isolate a procedural failure mode and establish its role in group-level misalignment within RLHF. The controlled setting isolates procedural effects but limits generalizability beyond this diagnostic context. This study uses controlled preference data, fixed clustering assumptions, and intervenes only at the reward learning stage. These choices clarify aggregation's role in group-level misalignment but defer questions of mode interpretability, adaptive clustering, and end-to-end deployment dynamics to future work. Future work should extend these findings to production preference datasets and motivate procedural audits of alignment pipelines prior to deployment, particularly for agentic systems whose behavior compounds over time.

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
