# OpenReview forum: "Procedural Fairness Failures in RLHF from Preference Averaging"
_ICLR.cc/2026/Workshop/AFAA — AFAA 2026 Poster_

### Official Review · Reviewer_AvsF · 2026-02-18
**Procedural Fairness Failures in RLHF from Preference Averaging**

**Rating:** 2
**Confidence:** 4

**Summary:**

This paper claims standard RLHF can be *procedurally unfair* when human preferences are heterogeneous. The key idea is that RLHF typically trains **one reward model** from **aggregated** preference data, which implicitly “averages” competing preference styles. When one preference style is more prevalent in the dataset, it dominates reward learning, and **minority preference modes** end up under-optimized.

To address this, the authors propose **Preference-Aware RLHF (PA-RLHF)**, which changes only the **reward learning** stage. They embed prompt–response pairs, cluster them into **k=3** preference modes (described as concise / detailed / balanced), and train a **separate reward model per mode**, so each preference mode gets its own optimization signal. They evaluate by checking whether the mode-specific reward model ranks the preferred response higher (alignment accuracy) and report improvements in overall accuracy and a reduced best–worst group gap.

**Strengths:**

* Attempts to directly address **minority preference groups** and procedural fairness in RLHF.

**Weaknesses:**

* **Low to none reproducibility / verifiability**: No model details, hyperparameters, training set specifics beyond high-level counts, and no dataset access, which makes it hard to validate the claims independently.
* **Insufficient dataset description**: Limited information about the prompt set and the demographics/composition of the 60 users; this matters because preference heterogeneity and “minority/majority” are inseparable from who the raters are and what they are judging.
* **Missing baselines / alternative formulations**: The paper doesn’t compare against simpler baselines like using an LLM to label preference modes (instead of clustering with k=3), or against a learned classifier for mode assignment. Without these, it’s unclear whether the gains come from “procedural fairness” per se or from a particular clustering choice.
* **Why not train a classifier?** If mode membership is stable, a supervised (or weakly supervised) classifier could replace clustering; the paper doesn’t explore this or explain why clustering is preferable.
* **No downstream impact demonstrated**: The evaluation stays at reward-model ranking accuracy; there’s no end-to-end RLHF policy training or demonstration that this improves downstream task behavior, user satisfaction, or agentic performance. This makes it difficult to assess practical significance beyond the diagnostic setup.
* **Fixed k=3 feels arbitrary**: Even if motivated by “stable structure,” the lack of sensitivity analysis for k (and robustness to clustering errors) weakens the generality of the approach.

---

### Official Review · Reviewer_c293 · 2026-02-18
**Promising Fairness Approach in RLHF with Limited Empirical Validation**

**Rating:** 3
**Confidence:** 4

**Summary:**

This work introduces preference averaging via their PA-RLHF framework. By collapsing diverse human preferences into a single reward model, standard RLHF causes majority groups to dominate while systematically under-representing minority views. Their  PA-RLHF framework trains separately different modes to ensure procedural fairness.

**Strengths:**

- Results. Preliminary results suggest that PA-RLHF can reduce fairness gaps compared to standard approaches.
- Problem Statement. This work aims to investigate how preference aggregation is done in RLHF, a very timely and relevant fairness issue.

**Weaknesses:**

- Results. The empirical results are limited in scope. As acknowledged by the authors, the evaluation relies on preliminary node choices, and experiments across datasets with different nodes would strengthen the claims.
- Formatting. The paper would benefit from clearer research questions and a more explicit statement of contributions.
- Datasets. More information about the 971 instances would help make the evaluation and reproducibility less difficult.
- Metrics. More extensive results could include more metrics than accuracy.

---

### Meta-Review · Area_Chair_fS3o · 2026-02-28

**Recommendation:** Tiny/Short Papers Track
**Confidence:** 3

**Metareview:**

This paper studies whether standard RLHF unfairly averages different human preferences and proposes PA-RLHF to model separate preference groups for better fairness. The idea is timely and relevant, especially for protecting minority preferences.

Both reviewers appreciate the motivation, but raise concerns about limited experiments, unclear explanations, and gaps in experimental rigor. Both reviewers agree that the paper tackles an important problem, how RLHF can unfairly marginalise minority preferences and find the preliminary results promising. However, they note major issues: unclear dataset details, limited metrics, missing baselines, arbitrary design choices and no downstream evaluation to show real-world impact. Reviewer 1 sees the direction as interesting but underdeveloped, while Reviewer 2 questions whether the improvements reflect fairness or just clustering.
Overall, the idea is valuable, but the experiments and reporting need much stronger support and clarity.

---

### Decision · Program_Chairs · 2026-03-02

Accept (Poster)